# The Efficacy of a Mix of Probiotics (*Limosilactobacillus reuteri* LMG P-27481 and *Lacticaseibacillus rhamnosus* GG ATCC 53103) in Preventing Antibiotic-Associated Diarrhea and *Clostridium difficile* Infection in Hospitalized Patients: Single-Center, Open-Label, Randomized Trial

**DOI:** 10.3390/microorganisms12010198

**Published:** 2024-01-18

**Authors:** Angela Saviano, Carmine Petruzziello, Clelia Cancro, Noemi Macerola, Anna Petti, Eugenia Nuzzo, Alessio Migneco, Veronica Ojetti

**Affiliations:** 1Emergency Medicine Department, Polyclinic A. Gemelli Hospital, 00168 Rome, Italy; angela.saviano@policlinicogemelli.it (A.S.); alessio.migneco@policlinicogemelli.it (A.M.); 2Internal and Emergency Medicine Department, Catholic University of the Sacred Heart, 00168 Rome, Italy; clelia.cancro01@unicatt.it; 3Internal Medicine Department, San Carlo di Nancy Hospital, 00165 Rome, Italy; cpetruzziello@gvmnet.it (C.P.); nmacerola@gvmnet.it (N.M.); apetti@gvmnet.it (A.P.); enuzzo@gvmnet.it (E.N.)

**Keywords:** *Limosilactobacillus reuteri* LMG P-27481, *Lacticaseibacillus rhamnosus* GG ATCC 53103, probiotic, antibiotic, diarrhea

## Abstract

Background: Antibiotic-associated diarrhea is a condition reported in 5–35% of patients treated with antibiotics, especially in older patients with comorbidities. In most cases, antibiotic-associated diarrhea is not associated with serious complications, but it can prolong hospitalization and provoke *Clostridium difficile* infection. An important role in the prevention of antibiotic-associated diarrhea is carried out by some probiotic strains such as *Lactobacillus GG* or the yeast *Saccharomyces boulardii* that showed good efficacy and a significant reduction in antibiotic-associated diarrhea. Similarly, the *Limosilactobacillus reuteri* DSM 17938 showed significant benefits in acute diarrhea, reducing its duration and abdominal pain. Aim: The aim of this study was to test the efficacy of a mix of two probiotic strains (*Limosilactobacillus reuteri* LMG P-27481 and *Lacticaseibacillus rhamnosus GG* ATCC 53103; Reuterin GG^®^, NOOS, Italy), in association with antibiotics (compared to antibiotics used alone), in reducing antibiotic-associated diarrhea, clostridium difficile infection, and other gastrointestinal symptoms in adult hospitalized patients. Patients and methods: We enrolled 113 (49M/64F, mean age 69.58 ± 21.28 years) adult patients treated with antibiotics who were hospitalized at the Internal Medicine Department of the San Carlo di Nancy Hospital in Rome from January 2023 to September 2023. Patients were randomized to receive probiotics 1.4 g twice/day in addition with antibiotics (Reuterin GG^®^ group, total: 56 patients, 37F/19M, 67.16 ± 20.5 years old) or antibiotics only (control group, total: 57 patients, 27F/30 M, 71 ± 22 years old). Results: Patients treated with Reuterin GG^®^ showed a significant reduction in diarrhea and clostridium difficile infection. In particular, 28% (16/57) of patients in the control group presented with diarrhea during treatment, compared with 11% (6/56) in the probiotic group (*p* < 0.05). Interestingly, 7/57 (11%) of patients treated only with antibiotics developed clostridium difficile infection compared to 0% in the probiotic group (*p* < 0.01). Finally, 9% (5/57) of patients in the control group presented with vomiting compared with 2% (1/56) in the probiotic group (*p* < 0.05). Conclusions: Our study showed, for the first time, the efficacy of these two specific probiotic strains in preventing antibiotic-associated diarrhea and clostridium difficile infection in adult hospitalized patients treated with antibiotic therapy. This result allows us to hypothesize that the use of specific probiotic strains during antibiotic therapy can prevent dysbiosis and subsequent antibiotic-associated diarrhea and clostridium difficile infection, thus resulting in both patient and economic health care benefits.

## 1. Introduction

Antibiotic-associated diarrhea (AAD) is defined as the presence of at least three soft or liquid stools per day for at least 24 h, during antibiotic treatment or in the two months following discontinuation that cannot be explained by other causes [1]. AAD has been reported in 5–35% of patients treated with antibiotics, especially in hospitalized patients compared to outpatients [1,2]. In most cases, AAD was not severe, but it can prolong hospitalization and may be associated with *Clostridium difficile* infection (CDI) [3,4]. Antibiotic use is the most common and significant cause of dysbiosis, leading to both short- and long-term changes in the gut microbiota. After a few days of antibiotic treatment, some important effects on the lumen and mucosal gut bacteria occur, such as a reduction in taxonomic richness and significant upregulation of genes that confer resistance to some classes of drugs [5]. Antibiotic-associated dysbiosis can last up to six months. It usually manifests as dyspepsia, abdominal pain, and diarrhea. However, more severe complications can occur, especially in older patients and in patients with multiple comorbidities leading to prolonged hospitalization and the increased prescription of diagnostic tests and therapies, with increased healthcare costs. In addition, the occurrence of gastrointestinal symptoms leads patients to discontinue therapy or take it at the wrong dosage, promoting the onset of antibiotic resistance [6,7]. An antibiotic’s effect on the gut microbiota depends on its spectrum of action, dosage, pharmacokinetics, and duration of treatment [8]. A high dosage and duration of therapy results in a stronger impact with increased dysbiosis [9]. Oral antibiotics that are well absorbed in the small intestine have less influence on the microbial ecosystem of the large intestine, while those that are poorly absorbed can cause significant changes. Intestinal dysbiosis has serious implications on host health: susceptibility to pathogens is increased due to reduced resistance of the commensal flora to colonization, and CDI may occur [10,11]. Dysbiosis of the gut microbiota can also lead to decreased short-chain fatty acid (SCFA) production, electrolyte imbalance, and AAD. Antibiotics can also lead to gut barrier dysfunction as they disrupt intestinal tight junctions and activate the NLRP3 inflammasome and autophagy in the gut [12,13,14,15,16]. The return to a normal composition of the gut microbiota after antibiotic treatment is often incomplete, and some species will never recover [17]. The factors that determine the recovery of the original microbiota or the transition to alternative equilibrium states are not yet clear; responses to broad-spectrum antimicrobial therapy are highly individualized and may vary after repeated exposure. A commonly used antibiotic in clinical practice is amoxicillin, which in combination with clavulanic acid, has devastating effects on the intestinal microbiota [18]. In the literature, a reduction in Gram-positive aerobic cocci and increased resistance of enterobacteria were observed. Fecal samples from patients with AAD were analyzed before and after treatment with amoxicillin–clavulanate. Four days after antibiotic treatment, no Bifidobacteria and Clostridia of cluster XIVa were detected, but a significant increase in *Enterobacteriaceae* was observed [19]. After the end of treatment, a reversal of these changes was observed, with the exception of *Bifidobacterium*, which was no longer found [20]. Oral treatment with vancomycin also leads to a significant reduction in several Enterococcus species as well as Clostridia, *Bifidobacteria*, and *Bacteroides* spp., and promotes the growth of less susceptible *Enterococci*, *Lactobacillaceae*, and potentially pathogenic *Enterobacteriaceae* [17,21,22]. Third-generation cephalosporins, such as ceftriaxone, promotes the proliferation of *Enterococci* and *Candida* spp., as well as CDI. An important role in the prevention of AAD is played by some probiotic strains such as *Lactobacillus* GG (*L. GG*) and the yeast *Saccharomyces boulardii* (*S. boulardii*), which have shown the best efficacy by demonstrating a significant reduction in the incidence of AAD [23,24]. Similarly, *Limosilactobacillus reuteri* (*L. reuteri*) DSM 17938 showed significant benefit in acute diarrhea by shortening its duration, as well as in infants’ abdominal colic and functional abdominal pain in infants. A new strain of *L. reuteri* (LMG P-27481) also showed beneficial properties; furthermore, it showed inhibitory effects on other pathogens such as *Escherichia coli*, *Salmonella*, and *Rotavirus* [25,26]. The aim of our study was to investigate the safety, tolerability, and efficacy of a probiotic combination (*Limosilactobacillus reuteri* LMG P-27481 and *Lacticaseibacillus* GG ATCC 53103) in the prevention of AAD, CDI, and other gastrointestinal symptoms in hospitalized adult patients receiving antibiotic therapy. The investigated product is commercially available as Reuterin^®^ GG in the form of a 1.4 g oral stick formulation with a probiotic concentration of *Limosilactobacillus reuteri* LMG P-27481 and *Lacticaseibacillus* GG ATCC 53103 of 2 × 10^10^ colony-forming units (CFUs). This probiotic combination was administered as an additional treatment to antibiotic therapy.

## 2. Endpoints of the Study

The primary endpoint of the study was to assess the incidence of AAD and CDI in the group receiving the combination of probiotics plus antibiotic therapy compared to the control group (antibiotic therapy alone). The secondary endpoints were the assessment of the occurrence of clinical gastrointestinal symptoms (vomiting and abdominal pain), the reduction in the number of daily bowel movements, the need for medication or other treatments, and the occurrence of adverse events in the group treated with probiotics compared to the control group.

## 3. Patients and Methods

This single-center, randomized, open-label study involved 113 (49M/64F, mean age 69.58 ± 21.28 years) adult patients treated with antibiotics from January 2023 to September 2023, who were hospitalized at the Department of Internal Medicine of the San Carlo di Nancy Hospital in Rome. The study was approved by the Ethics Committee of the San Camillo Forlanini Hospital, Protocol n. 343/CE Lazio 1. The study was conducted according to the Declaration of Helsinki. Patients receive no payment for their participation in the study.

### 3.1. Inclusion and Exclusion Criteria

Inclusion criteria were:-Age > 18 years-Patients who provided signed informed consent to participate in the study-Proven or suspected bacterial infection with a need for antibiotic therapy-Oral or parenteral antibiotic administration for a duration of at least 5 days

Exclusion criteria were:-Age < 18 years-Sepsis or severe generalized infection with shock-Acute pancreatitis-Known immunodeficiency or chronic gastrointestinal disease-Oncology patients undergoing active chemotherapy-Severe neurological pathology or inability to verbalize-Enteral or parenteral nutrition-Gastrointestinal malformation or gastrointestinal or abdominal surgery-Acute or chronic diarrhea already present at the start of antibiotic therapy-Use of any probiotic at enrollment or within the previous four weeks-Antibiotic therapy started more than 24 h before enrollment-No signed informed consent

### 3.2. Randomization

A computer system generated the randomization of enrolled patients using blocks of two, with an allocation ratio of 1:1. Group A received Reuterin^®^ GG (Noos, Rome, Italy) (BCCMTM Bacterial Collection of Ghent, Belgium, Italian Patent: 102016000011071, International Patent Request: PCT/IB 2017/053856) at the start of antibiotic therapy as an adjunctive treatment. The type, dose, and duration of antibiotic treatment was decided by the physician using the principles of good clinical practice and evidence-based medicine. We administered the first dose of probiotics on the same day (within the first 24 h) at the start of antibiotic therapy. In the following days and throughout the duration of antibiotic therapy, the product was taken as an oral stick on an empty stomach twice a day, in the morning and evening, and stopped with antibiotics. Group B (control group) was treated with antibiotic therapy alone. 

### 3.3. Statistical Sample Calculation

Based on data available in the literature, AAD (see previous paragraphs) has an adjusted incidence of 0.30. Studies in the literature have shown a 70% reduction in AAD in children who were taking probiotics [15]. To estimate the prevalence of AAD with a precision of 0.05 (5%) and power of 80%, a sample size of 110 patients was calculated, with 55 patients in each group.

## 4. Statistical Analysis

Data were collected in a database and analyzed using STAT-14^®^ for Mac statistical software. The characteristics of the patient sample were described using descriptive statistical analysis techniques. Quantitative variables were summarized through their mean ± standard deviation and with 95% confidence intervals; qualitative variables were presented through frequency tables (absolute and percentage). Comparisons were made by employing parametric and non-parametric statistical tests, depending on the type of variables under consideration. Values of *p* < 0.05 were considered significant. Data from patients with characteristics compatible with the above inclusion criteria were collected. 

## 5. Methods

We collected data regarding age; sex; anthropometrics; vital parameters; site of infection; laboratory tests performed; type, dose, and mode of antibiotic administration; previous AAD or other gastrointestinal symptoms; bowel movement; the presence of a family history of AAD or gastrointestinal disorders; comorbidities; and associated treatment for each patient using the first form. The second form was a daily diary completed by the patient in which he or she recorded the frequency of evacuations; stool consistency (assessed by the Bristol scale); intensity of any abdominal pain (assessed by the NRS scale); presence and duration of fever, vomiting, or other symptom; and general well-being (assessed on a scale of 1 to 10). The diary was completed daily throughout the duration of antibiotic therapy administration. In cases of diarrhea, stool tests for CDI toxin were performed. AAD was considered in cases of unexplained diarrhea, stool test negativity, or the presence of CDI toxin, occurring from the start of antibiotic therapy until 4 weeks after the end of antibiotic therapy. Abdominal pain was considered whenever pain was located in the abdomen, regardless of its duration. Pain intensity was quantified on a scale from 1 to 10 (the NRS scale). Constipation was considered at <3 evacuations per week or in the presence of an evacuation of hard stools (Bristol scale score 1 or 2). Patients who developed diarrhea dropped out of the study and continued or started probiotic supplementation, combined with fasting and specific antibiotic therapy if necessary. Patients who tested positive for CDI received specific therapy with vancomycin or metronidazole. After 4 weeks after discharge, all patients were re-contacted by telephone to complete follow-up. All patients were instructed by the doctor that this mix of probiotics could help reduce AAD. Protocol adherence was verified through the sachet count in the boxes. 

## 6. Results

A total of 113 patients (49M/64F, mean age 69.58 ± 21.28 years) enrolled in this study were randomized into Group A (Reuterin GG^®^), which comprised 56 patients (19M/37 F, mean age 67.16 ± 20.56), and Group B (control group), which comprised 57 patients (30M/27 F, mean age 71.9 ± 22.6) [Figure 1]. There were no statistically significant differences regarding age, gender, or comorbidities among patients in both groups. The type, dose, and duration of antibiotic treatment were decided according to the diagnosis of admission. In particular, the most common infections recorded were pulmonary infection, urinary tract infection, and biliary infection [Table 1]. The majority of patients had community acquired infections and only 17% (19/113) had hospital-acquired infections, without any difference between the two groups. All patients recruited completed the study. We observed no side effects associated with the use of the probiotics. The demographic characteristics of the patients are shown in Table 1. The types of antibiotics used are shown in Table 2. The most common antibiotic prescribed was ceftriaxone in 45% of patients and piperacillin/tazobactam in 30% of patients. Most of our patients were treated with a single antibiotic. We did not observe any difference between the two groups with regard to the type of antibiotics administered or the combination of antibiotics, as well as the average duration of treatment [Table 2]. The mean duration of antibiotic treatment was 10 days.

### 6.1. Antibiotic-Associated Diarrhea

None of the patients presented with diarrhea at the time of enrollment. A total of 16 out of 57 (28%) of the control group developed AAD during antibiotic therapy, compared with 11% (6/56) in the probiotic group (*p* < 0.01) [Figure 2]. AAD developed after four days of treatment. None of the patients during the follow-up developed AAD. Patients who developed diarrhea were considered to have dropped out and started probiotic therapy.

### 6.2. Clostridium Difficile Infection (CDI)

Interestingly, any patient treated with antibiotics plus the supplementation of these two specific probiotics developed CDI vs. 7/57 (11%) of patients treated only with antibiotics (*p* < 0.01) [Figure 2]. The development of CDI occurred after a mean of seven days after the start of antibiotic therapy. Based on the diagnosis, CDI patients started a specific antibiotic therapy according to guidelines suggested.

### 6.3. Gastrointestinal Manifestations: Diarrhea and Vomiting

At enrollment, none of the patients in the two groups showed gastrointestinal symptoms such as diarrhea or vomiting. After antibiotic treatment, 9% (5/57) of patients in the control group presented with vomiting, compared with only 2% (1/56) in the Reuterin GG^®^ group (*p* < 0.05) [Figure 2].

### 6.4. Abdominal Pain and the Number of Daily Bowel Movements

With regard to abdominal pain, we did not observe any difference between the two groups both at enrollment and at the end of therapy. No adverse events were reported with the use of probiotics.

## 7. Discussion

Our study showed that supplementation with this mix of probiotics (*Limosilactobacillus reuteri LMG P-27481* and *Lacticaseibacillus rhamnosus GG ATCC 53103*, Reuterin GG^®^) during antibiotic treatment is able to significantly reduce the development of AAD and CDI (*p* < 0.01). Other studies in the literature have compared the addition of probiotics to antibiotic treatments, showing that some probiotic bacteria, such as *Lactobacillus*, *Enterococcus*, *Bifidobacterium*, *Streptococcus*, *Saccharomyces*, and/or *Bacillus*, alone or in combination, may reduce the risk of AAD [27]. Probiotics, in fact, can improve the function of both the gut barrier and the gut mucosa. They can act at multiple levels, for example, in signaling pathways, in increasing the mucus layer, and in the production of defensins and tight junctions. Furthermore, they can compete for binding sites and inhibit the adhesion of pathogens [28]. Probiotics also have a role in the modulation of the innate and adaptive immune systems, in the promotion of antimicrobial factors and immunoglobulin A, and in the activation of intraepithelial immune cells (such as intraepithelial lymphocytes and other adaptive immune cells) against external microorganisms [29,30,31]. Further, as revealed by our study, probiotics can prevent the condition of AAD, which is responsible for the high rate of hospitalization and increased healthcare system costs. In two thirds of cases of AAD, the etiology is unknown, but in the remaining one third of cases, *Clostridium difficile* has been identified as the most common pathogen. AAD can be classified into two groups: early-onset, when diarrhea occurs during antibiotic treatment, and late-onset, when it appears 2–8 weeks after antibiotic withdrawal. The main mechanism underlying the pathogenesis of AAD is recognized in the condition of dysbiosis of the gut microbiota due to antibiotic treatment, both in terms of the overgrowth of pathogenic microorganisms and in terms of altered metabolic activity [32,33]. Antibiotic administration can promote pathogen overgrowth; alter metabolic activity, particularly the digestion and fermentation of carbohydrates in the colon; and lead to SCFA deficiency with voluminous osmotic diarrhea (>800–1000 g/day) that will persist despite fasting. In addition, some antibiotics cause thinning of the mucosal layer and disruption of tight junctions, making the intestinal epithelium more susceptible to damage and causing an increase in leaky gut and intestinal permeability [34,35]. Many clinical and experimental data indicate the beneficial role of probiotics, even if the efficacy of probiotics is often strictly strain-specific and limited to specific disease conditions. This implies that newly isolated probiotic strains should be targeted for specific diseases. Different factors can determine the efficacy of probiotic products in specific therapeutic contexts and according to specific individual differences, for example, the presence of comorbidities, drugs taken chronically, genetic factors, age, and gut microbiome composition. Studies in the literature have focused largely on bacteria such as *Lactobacillus* and *Bifidobacterium* or the yeast *Saccharomyces boulardii*. In addition, other recently discovered bacterial strains, such as *A. muciniphila* and *Faecalibacterium prausnitzii*, showed beneficial effects such as the reinforcement of the gut barrier, a decrease in inflammation, and improvement of the immune system. As mentioned previously, antibiotics are also known risk factors for CDI, which can be complicated by pseudomembranous colitis and toxic megacolon that can be potentially fatal [4,36,37]. Probiotics may offer an alternative and prospective strategy for the prevention and treatment of CDI [38,39,40]. Potential beneficial effects against CDI have been reported by *Lactobacillus, Saccharomyces*, and *Bifidobacterium* [41,42]. In a previous published study, it was demonstrated both in vitro and in vivo that *L. reuteri* LMG P 27481 in particular reduces the *C. difficile* DNA concentration in the caecum and *C. difficile* toxin titers in the gut lumen [43]. A systematic review and meta-analysis of 31 randomized controlled trials of 8672 patients suggested that probiotics are effective for preventing CDI [44]. Despite the heterogeneity of patients, most studies indicate that probiotics are effective in preventing AAD and CDI. Probiotics appear to be safe and effective (with rare side effects in immunocompromised or severely debilitated patients) [45,46,47,48,49]. In our study, we used a combination of two probiotic strains (*Limosilactobacillus reuteri LMG P-27481* and *Lacticaseibacillus rhamnosus GG ATCC 53103)*, commercially available as Reuterin GG^®^, in a 1.4-g oral stick formulation with a probiotic concentration of 2 × 10^10^ colony-forming units (CFUs)*. Lacticaseibacillus rhamnosus* GG is one of the most studied probiotic strains with proven effectiveness in reducing the incidence of AAD and other gastrointestinal disorders. *Lacticaseibacillus GG* ATCC 53103 has been studied in both adult and pediatric populations. A study by Francavilla et al. [42] showed that in children with AAD, *Lacticaseibacillus GG* (at a dosage of 3 × 10^9^ CFU, twice daily) was effective at reducing diarrhea and abdominal pain by restoring normal intestinal permeability. It is currently indicated in the treatment and prevention of AAD, so much so, that it is also recommended by the European Society of Pediatric Gastroenterology and Nutrition (ESPGHAN) at a dosage of at least 10^9^ CFU/day during the hospitalization period [42,44]. In our study too, we found that only 11% (11/56) of patients in group A (Reuterin GG^®^) developed diarrhea compared to 28% (16/57) of patients in the control group, who were not treated with probiotics. Further, 9% (5/57) of patients in group B presented with vomiting, compared with only 2% (1/56) in the Reuterin GG^®^ group. In our study, patients treated with only antibiotics had a higher incidence of CDI compared with patients who consumed probiotics in addition to antibiotics. A total of 11% (7/57) of patients in group B developed CDI, compared with 0% (0/56) in group A. A recent study by Sagheddu et al. [50] showed that *L. reuteri LMG P-27481*, thanks to its ability to adhere to human enterocytes and to stimulate the secretion of anti-inflammatory cytokines, represents a promising probiotic candidate for the prevention of antibiotic-induced diarrhea and CDI. Probiotics can be considered safe, with few side effects. Patients can start probiotics on the first day of antibiotic therapy and continue some weeks after completing antibiotic therapy. Shen [43] conducted a meta-analysis showing that probiotics may be more effective in reducing the risk of CDI and AAD when administered during the entire period and commenced closer to the first antibiotic dose. In our study, we started probiotics within the first 24 h from the first antibiotic administration. A review of probiotics for the prevention of AAD in children (23 studies with >3500 participants) reported that children treated with probiotics (*Lacticaseibacillus rhamnosus* or *Saccharomyces boulardii* at 5 to 40 billion CFU/day) were less likely to have AAD compared to the control group [44]. A trial of approximately 300 hospitalized children treated with antibiotics and *Saccharomyces boulardii* showed a reduced duration of diarrhea and lower stool frequency [48]. Another meta-analysis of sixty randomized controlled trials (RCTs) with more than 10,000 children and adults that compared probiotic treatments with placebo showed a significant reduction in the risk of AAD and CDI [49]. With regard to the adult population, our study added important results showing that specific probiotic strains (*Limosilactobacillus reuteri* LMG P-27481 and *Lacticaseibacillus rhamnosus GG* ATCC 53103*)* are effective in preventing AAD and CDI. However, other studies are needed to explore this field. Our study presents some limitations: first of all, it had a single-center and open-label design without a placebo control group, and with a small number of patients. However, it is well known that the degree of abdominal pain could be affected by a placebo result but the development of AAD or CDI are objective parameters, suggesting that there is a modification of the gut microbiota by the specific strains that we used. 

## 8. Conclusions

Our study showed, for the first time, the efficacy of two specific probiotic strains (*Limosilactobacillus reuteri LMG P-27481* and *Lacticaseibacillus rhamnosus GG ATCC 53103*) in preventing AAD and CDI in a population of 113 adult hospitalized patients treated with antibiotic therapy (probiotics were effective in reducing both the incidence of diarrhea, gastrointestinal symptoms, and CDI; *p* < 0.05). Based on these results, we hypothesize that the use of specific probiotic strains during antibiotic therapy can play an important role in preventing dysbiosis and subsequent AAD and CDI, with both patient and economic health care benefits (i.e., reduced days of hospitalization). More studies are needed to confirm these results and continue exploring this extensive field. 

## Figures and Tables

**Figure 1 microorganisms-12-00198-f001:**
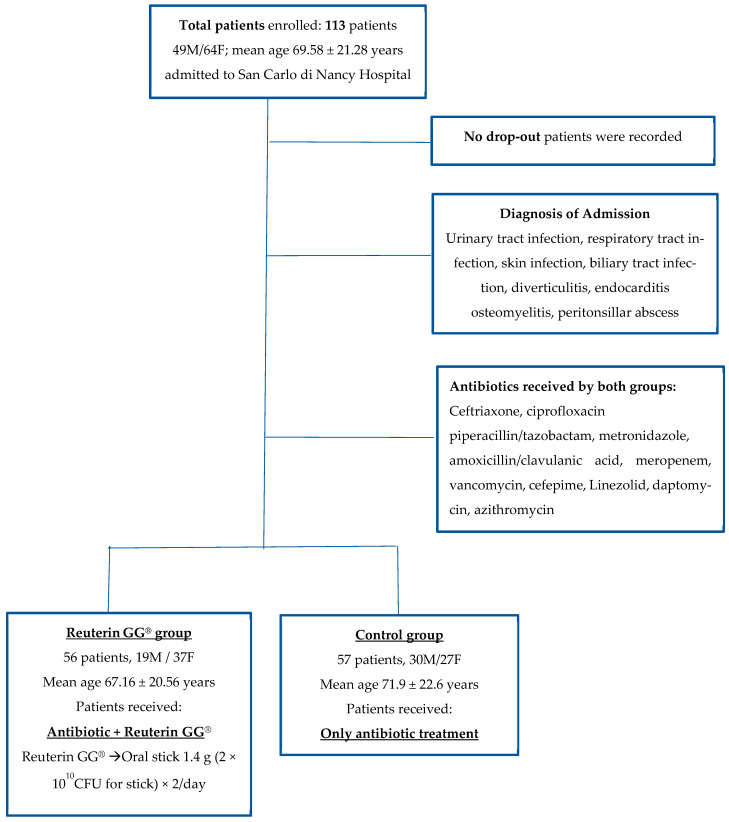
Flow chart of enrolled patients.

**Figure 2 microorganisms-12-00198-f002:**
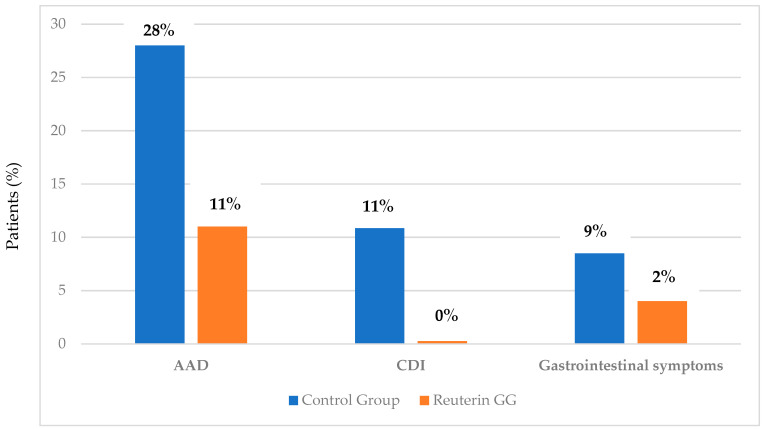
Evaluation of AAD, CDI, and gastrointestinal symptoms between Reuterin GG^®^ group and control group.

**Table 1 microorganisms-12-00198-t001:** Patient characteristics at baseline.

Total Patients = 113	Group A (Reuterin GG®) = 56	Group B (Control Group) = 57	*p*-Value
F/M	37/19	27/30	ns
Mean Age ± DS	67.16 ± 20.56	71.9 ± 22.6	ns
Comorbidities (Y/N)	33 (58.9%)/23 (41%)	42 (73.6%)/15 (26.3%)	ns
Diabetes	13 (23.2%)	8 (14%)	
Hypertension	30 (53.5%)	37 (64.9%)	
Ischemic Cardiopathy	11 (19.6%)	15 (26.3%)	
Renal Failure	6(10.7%)	6(10.5%)	
COPD (Chronic Obstructive Pulmonary Disease)	7 (12.5%)	5 (8.7%)	
Diagnosis of Admission			ns
Urinary tract infection	11 (19.6%)	11 (19.2%)	
Respiratory tract infection	20 (35.7%)	9(15.7%)	
Skin infection	0 (0%)	2 (3.5%)	
Biliary tract infection	19 (33.9%)	19 (33.3%)	
Diverticulitis	4 (7.1%)	10 (17.5%)	
Endocarditis	2 (3.5%)	0 (0%)	
Osteomyelitis	0 (0%)	5 (8.7%)	
Peritonsillar Abscess	0 (0%)	1 (1.7%)	

**Table 2 microorganisms-12-00198-t002:** Antibiotic treatment.

Total Patients = 113 Type of Antibiotics	Group A (Reuterin GG^®^) = 56	Group B (Control Group) = 57	*p*-Value
Ceftriaxone	27 (48.2%)	26 (45.6%)	ns
Ciprofloxacin	1 (1.7%)	1 (1.7%)	
Piperacillin/Tazobactam	15 (26.7%)	18 (31.5%)	
Metronidazole	9 (16%)	8 (14%)	
Amoxicillin/Clavulanic Acid	1 (1.78%)	3 (5.2%)	
Meropenem	6 (10.7%)	7 (12.2%)	
Vancomycin	1 (1.7%)	9 (15.7%)	
Cefepime	0 (0%)	1 (1.7%)	
Linezolid	2 (3.5%)	2 (3.5%)	
Daptomycin	1 (1.7%)	1 (1.7%)	
Azithromycin	3 (5.3%)	1 (1.7%)	
Number of antibiotics			ns
1	45 (80.3%)	40 (70.1%)	
2	8 (14.2%)	14 (24.5%)	
3	3 (5.3%)	3 (5.2%)	
Days of antibiotic treatment	9	10	ns

## Data Availability

Data are contained within the article.

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
