# Peer review of "The Efficacy of a Mix of Probiotics (Limosilactobacillus reuteri LMG P-27481 and Lacticaseibacillus rhamnosus GG ATCC 53103) in Preventing Antibiotic-Associated Diarrhea and Clostridium difficile Infection in Hospitalized Patients: Single-Center, Open-Label, Randomized Trial"

_microorganisms, 2024, doi:10.3390/microorganisms12010198_

Round 1

Reviewer 1 Report

Comments and Suggestions for Authors

I read this manuscript and found it of certain interest. However, I think it is premature to conclude what the author stated:

1.     Title: Please add the type of study in the title

2.     Abstract: Please remove all abbreviations from the abstract.

3.     Introduction: Our study aimed to evaluate the safety, tolerability, and efficacy of a combination of probiotics (Lactobacillus Reuteri LMG P-27481 and Lactobacillus GG ACC 53103) in preventing AAD, CDI: Why did you choose these strains?

4.     The study methodology

a.     Is the trial registered at any international database? If yes, what is the trial registration number?

b.     The number of patients studied is relatively small; how was the sample size estimated?

c.      The flow chart diagram is not good please try to update it.

5.     Results: very good.

6.     Discussion: The Discussion lacks focus on the selected probiotic strains. The authors should concentrate on the interpretation of their findings and their relevance to the field of study.

7.     Limitations - there is no section describing the limitations of the study. These include small sample size, single-center, open-label, and confounders, among others. This will need to be included and addressed.

8.     Conclusion: Please consider revising the conclusion because, at this point, there's a need for a substantial amount of additional effort to thoroughly verify the conclusions being made.

9.     There are some grammatical and syntax errors in the manuscript. Please revise.

10.  The references need to be adjusted according to the journal style. 

Comments on the Quality of English Language

There are some grammatical and syntax errors in the manuscript. Please revise.

Author Response

REF 1

  1. Title: Please add the type of study in the title

We added

  1. Abstract:Please remove all abbreviations from the abstract.

We remove all abbreviations from the abstract.

  1. Introduction:Our study aimed to evaluate the safety, tolerability, and efficacy of a combination of probiotics (Lactobacillus Reuteri LMG P-27481 and Lactobacillus GG ACC 53103) in preventing AAD, CDI: Why did you choose these strains?

Many studies, as reported in the text, have shown beneficial effects in reducing AAD with both L GG and L Reuteri we therefore decided to use this new product that contains both and to validate them in vivo

  1. The study methodology
  2. Is the trial registered at any international database?

No

If yes, what is the trial registration number?

  1. The number of patients studied is relatively small; how was the sample size estimated?

Yes, we added this part with the sample side calculation.

  1. The flow chart diagram is not good please try to update it.

We modified

  1. Results: very good.
  2. Discussion: The Discussion lacks focus on the selected probiotic strains. The authors should concentrate on the interpretation of their findings and their relevance to the field of study.

We improved discussion as you suggested

  1. Limitations- there is no section describing the limitations of the study. These include small sample size, single-center, open-label, and confounders, among others. This will need to be included and addressed.

We included this part

  1. Conclusion: Please consider revising the conclusion because, at this point, there's a need for a substantial amount of additional effort to thoroughly verify the conclusions being made.

We revised conclusion

  1. There are some grammatical and syntax errors in the manuscript.Please revise.

We revised

  1.   The references need to be adjusted according to the journal style. 

We modified

  1. There are some grammatical and syntax errors in the manuscript.Please revis

We revised  the paper

Reviewer 2 Report

Comments and Suggestions for Authors

I have a few questions:

¾      The study included 113 hospitalized patients with bacterial infections. It would be interesting to know more about the patient profile, such as the type of infection, severity of disease, and other factors that could influence the risk of AAD. For example, were the patients hospitalized for community-acquired infections or hospital-acquired infections? Were the infections mild, moderate, or severe? Were the patients immunocompromised?

¾    The study found that the combination of Lactobacillus reuteri LMG P-27481 and Lactobacillus rhamnosus GG ATCC 53103 (Reuterin®GG) was effective in preventing AAD in patients receiving antibiotic therapy for at least 5 days. It would be interesting to know if the efficacy of probiotics is affected by the duration of antibiotic treatment. For example, are probiotics more effective at preventing AAD in patients who receive antibiotic therapy for 5 days than in patients who receive antibiotic therapy for 10 days?

¾    The study found that the combination of Lactobacillus reuteri LMG P-27481 and Lactobacillus rhamnosus GG ATCC 53103 (Reuterin®GG) was effective in preventing AAD at a dose of 2x1010 CFU per day. It would be interesting to know if higher or lower doses of probiotics are more effective in preventing AAD. For example, are probiotics more effective at preventing AAD at a dose of 3x1010 CFU per day than at a dose of 2x1010 CFU per day?

¾    The study found that the combination of Lactobacillus reuteri LMG P-27481 and Lactobacillus rhamnosus GG ATCC 53103 (Reuterin®GG) was effective in preventing AAD, but the study did not have a placebo group. It would be interesting to know if the efficacy of probiotics in preventing AAD is significantly higher than in the case of placebo. For example, would the incidence of AAD be 20% in the placebo group and 10% in the probiotic group?

After reading the discussion, the following questions come to mind:

¾    The study found that the combination of Lactobacillus reuteri LMG P-27481 and Lactobacillus rhamnosus GG ATCC 53103 (Reuterin®GG) was effective in preventing AAD and CDI. Why do you think these two strains were particularly effective?

¾    The study did not compare Reuterin®GG to other probiotic strains. What are some other probiotic strains that have been shown to be effective in preventing AAD and CDI?

¾    What are the potential mechanisms by which probiotics may protect against AAD and CDI?

¾    What are the long-term effects of taking probiotics? Are there any potential risks associated with long-term use?

¾    How can healthcare providers best educate patients about the benefits of probiotics and how to choose the right probiotic for them?

 The conclusions could be improved by being more specific about the mechanisms by which probiotics may protect against AAD and CDI. The conclusions could also be more cautious about the generalizability of the findings.

Author Response

Ref. 2

  1. The study included 113 hospitalized patients with bacterial infections. It would be interesting to know more about the patient profile, such as the type of infection, severity of disease, and other factors that could influence the risk of AAD. For example, were the patients hospitalized for community-acquired infections or hospital-acquired infections?

We added this data

  1. Were the infections mild, moderate, or severe?

It’s very difficult to define the severity of infection, but we exclude patients with severe infection who need ventilator support of intensive/sub-intensive support

  1. Were the patients immunocompromised?

None of the patients had HIV or active cancers nor were they under active chemotherapy.

  1. The study found that the combination of Lactobacillus reuteri LMG P-27481 and Lactobacillus rhamnosus GG ATCC 53103 (Reuterin® GG) was effective in preventing AAD in patients receiving antibiotic therapy for at least 5 days. It would be interesting to know if the efficacy of probiotics is affected by the duration of antibiotic treatment. For example, are probiotics more effective at preventing AAD in patients who receive antibiotic therapy for 5 days than in patients who receive antibiotic therapy for 10 days?

The intestinal damage and dysbiosis are strongly link to the dosage and duration of antibiotic therapy. So the prolonged antibiotic therapies are at higher risk to develop AAD

  1. The study found that the combination of Lactobacillus reuteri LMG P-27481 and Lactobacillus rhamnosus GG ATCC 53103 (Reuterin®GG) was effective in preventing AAD at a dose of 2x1010 CFU per day. It would be interesting to know if higher or lower doses of probiotics are more effective in preventing AAD. For example, are probiotics more effective at preventing AAD at a dose of 3x1010 CFU per day than at a dose of 2x1010 CFU per day?

Literature studies show that a daily intake of at least 5 × 109 CFU is associated with significant efficacy for AAD and that higher probiotic dose is linked to greater efficacy. Although only few dose-effect studies have been performed, they observed a positive correlation between dose and AAD. Moreover, it could be necessary to test the new concentration …but basing on ESPGHAN guidelines for the prevention of AAD an high dose of L. rhamnosus GG is recommended (1–2x1010 CFU). However, the total dose of 2x1010 CFU of Reuterin® GG considers the presence and efficacy of L. reuteri LMG P 27481.

  1. The study found that the combination of Lactobacillus reuteri LMG P-27481 and Lactobacillus rhamnosus GG ATCC 53103 (Reuterin®GG) was effective in preventing AAD, but the study did not have a placebo group.

 You are right this is a limitation that we added in the discussion.

  1. It would be interesting to know if the efficacy of probiotics in preventing AAD is significantly higher than in the case of placebo. For example, would the incidence of AAD be 20% in the placebo group and 10% in the probiotic group?

It is well known that the degree of abdominal pain is affected by a placebo result but the development of AAD or CDI, are objective parameter, suggesting that there is a modification of the gut microbiota by the specific strains we used.

  1. After reading the discussion, the following questions come to mind: The study found that the combination of Lactobacillus reuteri LMG P-27481 and Lactobacillus rhamnosus GG ATCC 53103 (Reuterin®GG) was effective in preventing AAD and CDI. Why do you think these two strains were particularly effective?

One of the most studied probiotic strains is Lactobacillus rhamnosus GG, which has been repeatedly proven effective in reducing in the incidence of diarrhea in antibiotic-treated patients and in treating other gastrointestinal disorders. In a previous published study, it was demonstrated both in vitro and in vivo that especially L. reuteri LMG P 27481 reduces C. difficile DNA concentration in caecum and C. difficile toxin titers in the gut lumen.

  1. The study did not compare Reuterin®GG to other probiotic strains. What are some other probiotic strains that have been shown to be effective in preventing AAD and CDI?

Potential beneficial effects against CDI have been reported by Lactobacillus, Saccharomyces, and Bifidobacterium.

  1. What are the potential mechanisms by which probiotics may protect against AAD and CDI?

We added in the discussion

  1. What are the long-term effects of taking probiotics? Are there any potential risks associated with long-term use?

No side effects have been reported.

There not any potential risks associated with long-term use especially in immunocompetent patients.

  1. How can healthcare providers best educate patients about the benefits of probiotics and how to choose the right probiotic for them?

Literature studies show that probiotics are strain specific for diseases. So, to improve benefits derived from probiotics assumption healthcare can choose based on strain specificity of them.

  1. The conclusions could be improved by being more specific about the mechanisms by which probiotics may protect against AAD and CDI. The conclusions could also be more cautious about the generalizability of the findings.

We modified conclusion.

Round 2

Reviewer 1 Report

Comments and Suggestions for Authors

The authors responded to all issues. I accepted the paper after the revision